# Esterase Activity of Serum Albumin Studied by ^1^H NMR Spectroscopy and Molecular Modelling

**DOI:** 10.3390/ijms221910593

**Published:** 2021-09-30

**Authors:** Daria A. Belinskaia, Polina A. Voronina, Mikhail A. Vovk, Vladimir I. Shmurak, Anastasia A. Batalova, Richard O. Jenkins, Nikolay V. Goncharov

**Affiliations:** 1Sechenov Institute of Evolutionary Physiology and Biochemistry, Russian Academy of Sciences, pr. Torez 44, 194223 St. Petersburg, Russia; p.a.voron@yandex.ru (P.A.V.); vladimir.shmurak@gmail.com (V.I.S.); batalova.phys@gmail.com (A.A.B.); ngoncharov@gmail.com (N.V.G.); 2Centre for Magnetic Resonance, St. Petersburg State University, Universitetskij pr., 26, Peterhof, 198504 St. Petersburg, Russia; m.vovk@spbu.ru; 3Leicester School of Allied Health Sciences, De Montfort University, The Gateway, Leicester LE1 9BH, UK; roj@dmu.ac.uk

**Keywords:** albumin, esterases, *p*-nitrophenyl acetate, *p*-nitophenyl propionate, NMR, molecular docking, molecular dynamics

## Abstract

Serum albumin possesses esterase and pseudo-esterase activities towards a number of endogenous and exogenous substrates, but the mechanism of interaction of various esters and other compounds with albumin is still unclear. In the present study, proton nuclear magnetic resonance (^1^H NMR) has been applied to the study of true esterase activity of albumin, using the example of bovine serum albumin (BSA) and *p*-nitrophenyl acetate (NPA). The site of BSA esterase activity was then determined using molecular modelling methods. According to the data obtained, the accumulation of acetate in the presence of BSA in the reaction mixture is much more intense as compared with the spontaneous hydrolysis of NPA, which indicates true esterase activity of albumin towards NPA. Similar results were obtained for *p*-nitophenyl propionate (NPP) as substrate. The rate of acetate and propionate release confirms the assumption that there is a site of true esterase activity in the albumin molecule, which is different from the site of the pseudo-esterase activity Sudlow II. The results of molecular modelling of BSA and NPA interaction make it possible to postulate that Sudlow site I is the site of true esterase activity of albumin.

## 1. Introduction

Albumin is the main protein in the blood of mammals, where its concentration is 500–700 μM. Albumin is able to bind almost all known drugs, many nutraceuticals and toxic substances. Three main sites for the interaction of albumin with ligands (Sudlow I, Sudlow II, and site III) have been identified (Figure 1), as well as several secondary sites. Due to the free thiol group of Cys34 (Figure 1), albumin can serve as a trap for reactive oxygen and nitrogen species, thus participating in redox processes. 

However, albumin is not only passive, but also an active participant in pharmaco- or toxicokinetic processes. Numerous experiments have shown pseudo-esterase (irreversible binding of the substrate to the protein) or esterase (binding of the substrate to the active site of the protein, followed by the dissociation of the complex into an enzyme and a product) activity of albumin towards: α-naphthyl acetate and *p*-nitrophenyl acetate (NPA), fatty acid esters, aspirin, ketoprofen glucuronide, cyclophosphamide, nicotinic acid esters, octanoylgrelin, nitroacetanilide, nitrotrifluoroacetanilide, organophosphorus compounds (OPs) [4]. Acetylation is a typical example of pseudo-esterase activity (pseudo first order reaction), when the decrease in the substrate is caused not by its hydrolysis, but by the formation of covalent bonds at many sites of the albumin molecule. NPA (a classical substrate of carboxylesterase, CES) is widely used to study the hydrolytic activity of albumin. The product of NPA hydrolysis, *p*-nitrophenol, has a yellow color with an absorption peak at a wavelength of 400-412 nm and can be easily detected by spectrophotometry. MALDI technique has revealed that acetylation of human serum albumin (HSA) by NPA occurs at 82 sites including 59 lysine residues (these are all lysines in the protein molecule), ten serines, eight threonines, four tyrosines, and one aspartate [5]. NPA shows the highest affinity for Tyr411 in Sudlow site II. Tyr411 is acetylated within the first 5 min of reaction of HSA with NPA [5]. However, adducts with lysines are more stable.

Esterase activity in albumin is mainly associated with acetylation and subsequent slow deacylation of Tyr411 [4,6]. The fact that the hydrolysis of esters, in particular NPA, by albumin is practically no different from the “classical” hydrolysis was first announced by Casida and Augustinsson in 1959 [7]. Studying the kinetics of NPA hydrolysis, they found that both reaction products, acetate and *p*-nitrophenol, are released in a ratio close to 1:1. Furthermore, after complete hydrolysis of one dose of NPA, the enzymatic activity of albumin remains and does not even decrease when the next dose of the substrate is added. The authors also pointed out that the kinetics of the ester hydrolysis reaction with albumin corresponds to Michaelis-Menten kinetics with the formation of an intermediate complex.

The existence of at least two different catalytic centers in the albumin molecule responsible for two types of activity, true- and pseudo-esterase, was proposed for the first time in 1972 [8]. The interaction of albumin with NPA has a biphasic character. During the first minutes, a “burst” of activity is observed, which means rapid formation of the reaction product *p*-nitrophenol. Then the system switches to a stationary mode, but does not reach a plateau. The first phase is provided by two processes at two different sites: monoacetylation of albumin as a result of pseudo-esterase activity of Sudlow site II and albumin-catalysed “true” hydrolysis at the second site. The second stage is the result of the activity of only the second site. These two phases are easily reproducible; a phase I inhibitory assay with NPA as a substrate was used by Zaidi et al. in 2013 [9]. It has also been shown that the hydrolysis of NPA results in acetylation of two tyrosines, but acetylation of the second tyrosine does not interfere with the catalytic activity of albumin [8]. The authors of the study proposed a working model, according to which the acetylation of the second tyrosine occurs near the site responsible for catalysis, and after the second tyrosine becomes acetylated, the acetate group goes directly to the water molecule. In the works of Spanish researchers under the supervision of E. Vilanova, it was found that albumin has catalytic activity towards OPs [10,11,12]. These reports unambiguously demonstrate the catalytic activity of albumin, and the kinetic constants for the studied substrates are given. The authors assign the main role in the detoxification of OPs by albumin to catalytic hydrolysis, while the auxiliary role to stoichiometric interaction with albumin. We were the first who suggested Sudlow site I to be the site of albumin true esterase activity towards OPs [13,14]. However, the mechanism of the catalytic activity of albumin is not fully understood and the existence of true esterase activity of albumin is still under discussion. 

Most of the studies on the kinetics of the interaction of albumin with NPA and similar substrates were based only on the kinetics of the release of *p*-nitrophenol, which could be the product of both esterase and pseudo-esterase activity of albumin. Thus, in our previous paper [15], we studied the kinetics of the interaction of NPA with albumins of different species measuring the yield of *p*-nitrophenol using spectrophotometric methodology. To distinguish the work of one site from another, kinetic constants were calculated at different intervals of the concentration curve: 0–60 s for the site of pseudoesterase activity Sudlow II and 90-600 s for the site of true esterase activity (suggested Sudlow I). The disadvantage of this approach is that even in the first seconds of the interaction of NPA with albumin, the site of true esterase activity already comes into operation, introducing distortions in the calculation of kinetic constants for the Sudlow site II.

It is possible to separate the kinetics of one activity from another by observing the yield of product of the true esterase reaction. In the case of NPA hydrolysis, it is the acetate moiety. One of the possible methods for the detection of acetate can be proton nuclear magnetic resonance (^1^H NMR). Molecular modelling methods could supplement the knowledge about the molecular mechanisms of the interaction of the substrate with the sites of esterase and pseudo-esterase activity of albumin. The purpose of the research reported on here was to apply NMR spectroscopy to detect the true esterase activity of bovine serum albumin (BSA) towards NPA. Then, using molecular modelling methods, to determine the site of protein esterase activity and reveal the details of NPA interaction with this site.

## 2. Results

### 2.1. True Esterase Activity of Albumin towards NPA According to ^1^H NMR

In the presented study, using ^1^H NMR technology, we have performed a qualitative analysis of the acetate release during the interaction of BSA with NPA. Figure 2 shows the dynamics of the spectrum of the mixture of BSA and NPA. Due to the features of NMR technology, usually 7–10 min elapse between the moments of mixing the substrate with the enzyme and recording of the first spectrum. By this time, Sudlow sites II of BSA molecules are already completely acetylated [5,15], therefore the obtained spectra reflect the kinetics of exclusively true esterase reaction of albumin. That is why such a delay is applicable for measurement of kinetics of albumin true esterase activity, however, to study the reaction of Sudlow site II (which occurs during the first 5 min of BSA-NPA interaction), faster NMR methods should be applied [16,17]. For example, rapid injection NMR spectroscopy allow the observation of fast chemical reactions and unstable intermediates [18,19].

According to the data obtained, signals c’ (7.90 ppm) and b’ (6.65 ppm) of *p*-nitrophenol (Figure 2A,B) and even the acetate signal a’ (1.85 ppm) (Figure 2A,D) are visible in the first spectrum already. It can be seen in Figure 2A–D, that within 50 min of observation, signals a’, b’, and c’ corresponding to the products of the esterase reaction are amplified, and signals a (2.27 ppm), b (7.25 ppm) and c (8.2 ppm) corresponding to NPA are attenuated. Impurities (3.59 and 1.12 ppm, marked with the asterisks in Figure 2A) are possibly related to the industrial synthesis of NPA.

Changes in the corresponding signals of NPA and the products of its hydrolysis in the absence of BSA (spontaneous hydrolysis) are shown in Appendix A. It can be seen even with the naked eye that in the case of spontaneous hydrolysis, the increase in the acetate signal intensity occurs much more slowly. At the next stage, to compare the kinetics of true esterase activity of albumin with the spontaneous hydrolysis of NPA, we plotted the dependence of the relative integral intensity of the acetate signal on time in the presence and in the absence of BSA in the reaction mixture (Figure 3).

According to the data obtained, the accumulation of acetate in the presence of BSA is much more intense as compared with the spontaneous hydrolysis of NPA. The data obtained indicate the presence of true esterase activity in albumin towards NPA. BSA content in a Sigma sample is claimed to be more than 96%. Hypothetically, other blood plasma esterases may be present in a commercial albumin sample in the remaining portion of less than 4%. It is known that CES exhibits high hydrolytic activity towards NPA [20]. However, in bovine blood (as well as in humans), in contrast to rodents, CES content is extremely low and is not detectable by common biochemical methods [21,22]. Therefore, the esterase activity revealed by us undoubtedly belongs to albumin. It is important to note that the active release of acetate is observed already in the first minutes of the interaction of BSA with NPA, while the half-life of acetylated Tyr411 is 61 ± 4 h [5]. As for adducts of acetate with other amino acids, their formation takes at least 30 min [5]. This result suggests that the albumin molecule has a site of true esterase activity, which is different from Sudlow site II. In the next step, using molecular modelling methods, we tried to identify this site.

### 2.2. Analysis of Possible Sites of Albumin Esterase Activity

As mentioned above, there are three main ligand-binding sites in albumin: Sudlow site I, Sudlow site II, and site III (Figure 1). According to numerous data, Sudlow site II with catalytic tyrosine Tyr411 is the main site of pseudo-esterase activity of the protein; therefore, enzymatic hydrolysis of NPA probably occurs in Sudlow site I or site III.

It is logical to assume that the catalytic amino acid that provides the reaction should not coincide with those amino acids that were irreversibly (or very slowly reversibly) acetylated in the study of Lockridge et al. [5]. Thus, all lysines are excluded, while serines, threonines and tyrosines remain. To find possible catalytic amino acids in Sudlow site I and site III, we have analysed crystal structures of complexes of HSA with warfarin in Sudlow site I (PDB structure code 2bxd [23]) and HSA with 4Z,15E-bilirubin-IXalpha site III (PDB structure code 2vue [24]) (Figure 4). Despite the fact that our NMR experiments are performed in BSA, we chose HSA crystal structures for analysis since warfarin and 4Z,15E-bilirubin-IXalpha are site-specific ligands. Moreover, they are large enough to occupy the whole sites and all serines, threonines and tyrosines of Sudlow site I and site III could be found for sure. According to crystallographic data, there are 7 binding sites for fatty acids (FA1-FA7) in albumin molecule [25]; Sudlow site I overlaps with FA7 and Sudlow site II overlaps with FA3 and FA4. We proposed that true esterase reaction occurs more likely in drug binding sites (Sudlow site I and site III) than in FA1-7, so we have not analysed albumin crystal structures encapsulating fatty acids.

Figure 4 shows all amino acids within 6 Å of warfarin (Figure 4A) and 4Z,15E-bilirubin-IXalpha (Figure 4B) sorbed in Sudlow site I and site III, respectively. Analysis of this environment has revealed Tyr150 and Ser287 in Sudlow site I, as well as Tyr138, Tyr161, and Ser193 in site III. Among these amino acids, Ser287 and Tyr161 are the sites of irreversible acetylation of HSA, while adducts of HSA with Tyr150, Tyr138 and Ser193 were not detected by MALDI-TOF [5]. These three residues are the main candidates to be the catalytic amino acids responsible for true esterase activity of albumin.

### 2.3. Molecular Docking of NPA into the Selected Binding Sites

Due to the deletion at position 116, amino acid numbering in BSA after amino acid residue 115 is shifted one position relative to HSA. Tyr138, Tyr150, Ser193, and Tyr411 in HSA correspond to Tyr137, Tyr149, Ser192, and Tyr410 in BSA. At the next step, we performed molecular docking of NPA into the suggested centers of esterase activity of BSA, as well as into the site of pseudo-esterase activity of the protein (Sudlow site II).

The key factor for the possibility of the esterase reaction is primarily the geometry of the productive complex. That is why, from all the found conformations of BSA-NPA complexes, we selected the conformations with the minimum distance between the carbonyl carbon of NPA and the hydroxyl oxygen atom of the catalytic amino acids (distC-O). For all four studied binding sites, such a conformation of NPA has been revealed, in which NPA is in the immediate vicinity of the catalytic amino acid (Figure 5).

In the case of Sudlow site I and Tyr149 (Figure 5A), the aromatic ring of NPA is located between the aromatic ring of Tyr149 and the imidazole ring of His241; the distC-O value is 0.40 nm. In the case of Sudlow site II and Tyr410 (Figure 5B), the aromatic ring and the nitro group of NPA bind surrounded by aliphatic amino acids Leu490, Val425, Phe487, Leu459; the value of distC-O is 0.37 nm. In the case of site III and Tyr137 (Figure 5C), the aromatic ring of the ligand binds surrounded by aliphatic amino acids Leu115, Ile 141 and Ile181, and the nitro group interacts with the side chain of Arg185. The value of distC-O is 0.40 nm in this complex. Finally, in the case of site III and Ser192 (Figure 5D), the aromatic ring of the ligand is bound surrounded by aliphatic amino acids Leu189 and Ala193. The nitro group interacts with the hydroxyl group of Ser428 and the side chain of Arg435. The value of distC-O is 0.37 nm.

At the next step, the stability of the obtained complexes was checked by molecular dynamics (MD) simulation.

### 2.4. Stability of NPA-BSA Complexes According to MD Simulation

At this stage, we checked the stability of the resulting complexes by short 10 ns MD simulation. The time dependences of distC-O values are shown in Figure 6.

According to the data obtained, only the complex of NPA with Sudlow site II remains stable for 10 ns (Figure 6B), which once again confirms the high reactivity of Tyr410 (Tyr411 in HSA). This is the first amino acid that NPA interacts with, and the stability of the productive conformation contributes to the rapid (relative to other amino acids of albumin) acetylation of the tyrosine. As for Sudlow site I, according to the data obtained, the orientation of NPA relative to the amino acids of the site remains unchanged most of the time (Figure 6A). The only exception is a short period (from 6 to 9 ns), but then the ligand molecule returns to its original conformation between Tyr149 and His241. In the case of Tyr137 (Figure 6C), NPA stays in the site for some time, then leaves it, and, in the final part of the simulation, interacts with Lys187, which is one of the sites of irreversible acetylation. In the case of Ser192 (Figure 6D), NPA immediately leaves the binding site and approaches Lys431 on the surface of the protein, which is also a site of irreversible acetylation.

Thus, it appears that Ser192 (Ser193 in HSA) is not the main site of albumin esterase activity. This assumption is supported by the fact that Ser192 of BSA (Ser193 of HSA) is replaced by alanine in rat serum albumin (RSA), while RSA and HSA show similar kinetics of NPA hydrolysis [15]. In the case of Tyr137 (Tyr138 in HSA), the NPA molecule also leaves the active site quite quickly, and we believe that this amino acid does not participate in esterase activity of the protein (at least it is not the main site). In the immediate environment of Tyr137, there are no amino acids capable of attracting the proton of catalytic tyrosine, in contrast to Sudlow site I, where His241 and His287 (His242 and His288 in HSA) are located near Tyr149. In our previous studies, we showed that warfarin (a specific ligand of Sudlow site I) inhibits the long-term stage of NPA hydrolysis by albumin [15]. Based on the totality of these data, we believe that it is Sudlow site I with catalytic Tyr149 (Tyr150 in HSA) that is the site of albumin true esterase activity.

### 2.5. Interaction of NPA with Sudlow Site I and Sudlow Site II of BSA According to MD Simulation

At the next stage, we studied the interaction of NPA with Sudlow sites I and II using MD over a longer trajectory (50 ns). The obtained trajectories were used to plot the dependence of the value of distC-O on time (Figure 7). According to the plots obtained, NPA remains inside Sudlow sites I and II throughout the entire simulation period; however, the position of the ligand near Tyr410 is more stable and more favorable for the nucleophilic attack of hydroxyl of catalytic Tyr to carbonyl carbon of NPA. In general, this is consistent with our previous biochemical experiments, which indicate that catalytic activity of Sudlow site I of BSA is four orders of magnitude lower than that of Sudlow site II [15].

It is interesting to note the role of His241 in the interaction of NPA with Sudlow site I. It could be assumed that the histidine plays a role similar to the role of histidine in the catalytic triad of serine hydrolases, which is attracting the proton of the hydroxyl group of Tyr149. However, according to MD simulation, the role of His241 is more likely to stabilise the position of NPA in the Sudlow site I. During 50 ns of simulation, the average distance between the carbonyl oxygen of NPA and atom Hε2 of histidine is 0.34 nm, while the distance between the hydrogen of the hydroxyl group of Tyr149 and atom Nδ1 of histidine is 1.02 nm. That is, at least in the case of NPA hydrolysis, His241 plays the role of an oxyanionic center.

In our previous studies, we simulated the interaction of paraoxon with Sudlow site I of BSA and HSA, and according to our data the hydroxyl of Tyr149 (Tyr150 in HSA) tended to interact with the imidazole ring of His241 (His242 in HSA). But at the same time, paraoxon moved away from Tyr149 (150) almost immediately, in the very first steps of MD [26,27]. Interestingly, according to biochemical data, warfarin (a specific inhibitor of Sudlow site I) inhibited the long-term stage of albumin-catalysed hydrolysis of NPA, but not of paraoxon [15], which indicates different mechanisms of NPA and paraoxon hydrolysis. In the case of NPA interaction with Sudlow site I, Asp258 can play the role of a proton attractant. In 50 ns of simulation, the distance between one of the oxygens of the side chain of Asp258 and the hydrogen atom of the hydroxyl group of Tyr149 decreases from 1.3 nm at the initial point of the trajectory to 0.2-0.5 nm in the last 10 ns of the trajectory (Appendix A).

Numerous experimental data indicate the important role of Arg409 (Arg410 in HSA) in pseudo-esterase activity of albumin in Sudlow site II. It is believed that this arginine acts as an oxyanionic center, forming a hydrogen bond with the carbonyl group of the substrate [28]. However, according to our data, NPA does not interact with Arg409 during the simulation. Since albumin is not a classical enzyme, and the hydrolysis reaction proceeds slowly (in comparison with acetylcholinesterase (AChE), which requires strong fixation of acetylcholine), we believe that there may be several productive conformations of the substrate, in which the attack of the hydroxyl group of the tyrosine on the carbonyl carbon is possible. In any case, Arg409 appears to be required for anchoring the substrate at the entrance of Sudlow site II. Nevertheless, according to our data, Sudlow site II of BSA is the most preferred site for NPA and Tyr410 is the most reactive amino acid, which is consistent with the literature data.

### 2.6. Interaction of NPA with Sudlow Site I and Sudlow Site II of Oxidised BSA According to MD Simulation

The albumin molecule contains one free thiol group within the Cys34 residue, which can form disulfides with cysteine and other blood plasma thiols or be oxidised to sulfenic and sulfinic acids [29], and oxidation might modulate the protein properties. In healthy people, about 70% of albumin remains in a reduced form, but the level of oxidised albumin can increase up to 70% in kidney and liver diseases or in other pathologies, as well as in athletes after intense physical training or during the aging process [30,31,32,33]. Moreover, several earlier studies have demonstrated that commercial albumin used in biochemical experiments can be oxidised: in different samples, the percentage of mercaptoalbumin (albumin with the reduced form of Cys34) varied from 12.6% to 48.03% [34,35,36]. 

Cysteine was found to be the major molecule attached to the sulfydryl group of the cysteinyl residue of serum albumin [37], therefore, in our study, we studied the interaction of cysteinylated BSA (cysBSA) with NPA.

The preparation of a model of cysBSA is described in Materials and Methods (Section 4.3). Since the structures of Sudlow sites I and II in the prepared models of reduced and oxidised BSA are identical (changes in the structure affected only Cys34 and its environment), molecular docking of NPA into cysBSA was not performed separately. For MD simulation of cysBSA-NPA complexes, we used the same starting coordinates of the ligand that were obtained as a result of molecular docking of NPA into Sudlow sites of the reduced BSA.

The time dependence of distC-O according to MD simulation is shown in Figure 8.

In the case of Sudlow site I (Figure 8A), NPA remains near the catalytic tyrosine only for 5 ns, then the ligand moves away, and after 30 ns it moves to another part of the site and approaches residues Arg198, Arg194 and Arg217, which form the entrance to Sudlow I. After NPA moves away from Tyr149, the latter approaches the imidazole ring of His241, while the interaction of Tyr149 with Asp258 does not occur (in contrast to reduced BSA). In the case of Sudlow site II (Figure 8B), NPA remains in the same conformation throughout the entire trajectory, the position of the ligand within the site is even more stable than in the case of the reduced protein. Earlier, we obtained a similar effect when simulating the interaction of paraoxon with Sudlow site II of HSA with different oxidation states of the thiol group [27]. Thus, according to our data, cysteinilation of the thiol group of Cys34 of BSA has little effect on the interaction of NPA with Sudlow site II. A similar result was obtained by Anraku et al. [38]. The authors oxidised human albumin *in vitro* by three different methods (by a metal–catalysed oxidation system (MCO), chloramine-T, and hydrogen peroxide) and studied how the modifications changed the functional properties of HSA including its pseudo-esterase activity towards NPA. Oxidation with hydrogen peroxide had practically no effect on the rate of hydrolysis, while the oxidation of MCO and chloramine-T slowed down the reaction. The different effect of different oxidants can be explained by the fact that MCO and chloramine-T can oxidise not only Cys34, but also the side chains of lysines and arginines, including Arg411 and Arg485 [38,39], localised in Sudlow site II. 

To understand how cysteinylation of the redox site of BSA affects Sudlow site I, we have drawn the web of interactions between the sites during MD simulation (Figure 9). According to the data obtained, oxidised Cys34 loses its connection with His39, disrupting the integrity of system Ser28-Cys34-His39-Tyr139. In oxidised BSA, Asp108 and His145 move towards Sudlow site I; Asp108 is captured by Tyr147, which is located between the redox site and Sudlow site I. We suppose that these events change the system of interactions between the sites, and Sudlow site I loses its affinity towards NPA.

Thus, Sudlow site I is more susceptible to allosteric modulation, which should be taken into account when developing drugs interacting with this site, especially when it comes to patients with chronic diseases accompanied by an increased level of oxidised albumin in the blood plasma.

### 2.7. Interaction of BSA with p-Nitrophenyl Propionate

To support our results on true esterase activity of albumin, we studied the interaction of BSA with *p*-nitrophenyl propionate (NPP) using NMR and molecular modelling methods. According to NMR spectroscopy, BSA possesses true esterase activity towards NPP too (Figure 10). During 91 min of observation, signals a_1_’ (0.98 ppm), a_2_’ (2.11 ppm), b’ (6.65 ppm) and c’ (7.90 ppm), corresponding to the products of the esterase reaction, are amplified; signals a_1_ (1.10 ppm), b (7.25 ppm) and c (8.20 ppm), corresponding to NPP, are weakened. Signal a_2_ corresponding to the CH_2_-group of NPP is not visible in the spectrum, since it is most likely overlapped by the BSA and DMSO signals (Appendix A). As in the case of NPA, the rate of spontaneous hydrolysis of NPP is lower than in the presence of BSA (Figure 10 and Appendix A).

Molecular docking of NPP into Sudlow site I (Figure 11A) and Sudlow site II (Figure 11B) of BSA has revealed that the ligand binds to these sites in a configuration close to the position of NPA molecule (Figure 5A,B). The values of distC-O are equal to 0.30 and 0.38 nm for Sudlow site I and Sudlow site II, respectively. The obtained result, as expected, indicates a similar mechanism of interaction of BSA with NPA and NPP.

According to our data, BSA hydrolyses NPP clearly slower than NPA (Figure 2 and Figure 10). However, for quantitative assessment of kinetic constants using NMR, the slow reaction rate is rather an advantage with respect to duration of scanning the sample at one time point. For the same reason, we used a reduced number of scans in order to find a compromise between the signal-to-noise ratio and the error in determining the hydrolysis rate due to prolonged scanning duration. A quantitative analysis of true esterase activity of albumin is the goal of our next studies.

## 3. Discussion

Obviously, one of the main reasons for the slow hydrolysis of NPA and other esters by albumin is the absence of a catalytic triad and an oxyanion center [28]. Catalytic triads are a widespread phenomenon, especially among hydrolases and transferases that provide covalent catalysis, the characteristic feature of which is the formation of a covalent intermediate product, which is then hydrolysed to regenerate the enzyme [40]. Not only in the case of albumin, but also in the case of “classical” serine hydrolases, the mechanism involves acylation and the sequential formation of two tetrahedral structures, and the second tetrahedron is formed with the participation of a water molecule and transacylation from a serine residue to a water molecule [28].

Even the cleavage of acetylcholine by AChE, which is one of the fastest enzymatic processes (k_cat_/K_M_ = 1.6×10^8^ M^−1^s^−1^ [41]), occurs according to the general scheme: the acylation stage proceeds according to the addition-elimination mechanism through the formation of a stable tetrahedral intermediate. During the nucleophilic attack of Ser203 of AChE to the carbonyl carbon of the substrate, protonation of His447 occurs, which plays the role of a base in this process. Protonated histidine is stabilised by interaction with Glu334. As a result, the formation of the first tetrahedral intermediate occurs at the acylation stage. It is stabilised by hydrogen bonds with backbone carbonyl oxygens of the residues of the oxyanion hole (Gly121, Gly122, Ala204). This intermediate is rapidly degraded by the action of protonated histidine, which acts as an acid, to an acyl enzyme with the release of choline [42]. In the very first articles describing the mechanism of action of AChE, the phenomenon of substrate inhibition of the enzyme was noted, the essence of which is inhibition of the deacetylation stage [43]. So there is nothing new and surprising in the formation of a covalent adduct (acyl enzyme), and the whole problem lies in the lifetime of the covalent bond, which depends on the structure and specialisation of the active center, substrate, pH and temperature.

It is axiomatic that the catalytic triad and the oxyanionic center as they exist in the AChE molecule are absent in the albumin molecule. According to our data, in the case of NPA hydrolysis by Sudlow site I of BSA, His241 can play the role of an oxyanion center, while Asp258 can attract the proton of the catalytic tyrosine. His241 in BSA corresponds to His242 in HSA and Asn242 in RSA. Asp258 in BSA corresponds to Asp259 in HSA and Glu259 in RSA. Both histidine and asparagine can act as an oxyanion center, while aspartate and glutamate can pull a proton away from tyrosine. Thus, the molecular modelling data do not contradict the experimentally derived fact that all three albumins have true esterase activity towards NPA. Earlier, based on modelling the interaction of albumin with OPs, we suggested that the catalytic “dyad” His-Tyr or Lys-Tyr, in which histidine or lysine acts as an acid residue, is required for the hydrolysis of some substrates with albumin [4]. Both cases are described in the literature: the sedolisin proteases carry out catalysis using the Ser/Glu/Asp triad, while the Ser/His/His catalytic triad was validated by the crystal structure of the cytomegalovirus protease [44]. The role of these amino acids in the hydrolysis of various substrates by albumin remains to be determined.

The hydrolytic activity of Sudlow site I of albumin has been described previously. Thus, it has been shown that Lys199 and Lys195 of HSA Sudlow I, along with other lysines of albumin, can be acetylated upon interaction with NPA. It is known that Lys199 can be acetylated when interacting with aspirin [45]. The mechanism of this interaction was determined by a method combining quantum and molecular mechanics (QM/MM) [46]. However, in these and similar works, we are still talking about irreversible acetylation, that is, about pseudoesterase activity. Among the works devoted to the interaction of albumin with aspirin, it is interesting to note the study of Honma et al. [47]. Back in 1991, using NMR technique, they registered the yield of the products of hydrolysis of aspirin with human albumin-acetate and salicylic acid. The authors could not observe the acetylated HSA and suggested that acetylated HSA was not formed at all during the reaction or that the compound is too unstable to be detected. On this basis, they concluded that the rate of deacetylation is much higher than the rate of acetylation of albumin, which means that is true esterase hydrolysis. Nevertheless, the work gives the dependence of the concentration on time only for salicylic acid (which can be a product of both pseudo- and true esterase activity), but not for acetate. The period of registration was as large as 80 h (!), which covers stages of both true esterase activity and extremely slow de-acetylation of acetylated albumin (pseudo-esterase activity). The catalytic amino acids that were responsible for detected hydrolytic activity were not discussed at all. Later, mass spectrometric methods revealed long-lived adducts of albumin with acetate after incubation of the protein with acetylsalicylic acid [45], therefore, the mechanism of interaction of aspirin with albumin remains open.

Dahiya et al. [48] studied the interaction of 3,5,6-trichloro-2-pyridinol (TCPy) and paraoxon methyl (PM) (metabolites of chlorpyrifos and methyl-parathion, respectively) with BSA by NMR and molecular modelling. According to the data obtained, PM, but not TCPy, can be hydrolysed in the presence of BSA faster than spontaneously. However, it should be noted that in order to calculate the kinetic constants, the authors were registering a decrease of the substrate but not the yield of the products; and again, an extremely long period of time was used for registration (the first point was at 12 h, and the total period was 120 h). The formation of both products was observed, but the calculation of the hydrolysis rate constants was carried out solely based on the decrease in the substrate, so they actually discussed the pseudo-first order reaction. Moreover, the difference between spontaneous and enzymatic hydrolysis of PM was as small as 12%, which was comparable to measurement error and cannot be regarded as a confident difference. Thus, we suppose that the study of Dahiya et al. [48] is again about pseudoesterase activity of albumin. At best, taking into consideration a very slow kinetics of hydrolysis of paraoxon as compared to NPA, they registered by NMR a total decline of the substrate as a result of its interaction with two or more albumin sites. In any case, the true esterase activity of albumin cannot be inferred from this approach and data obtained.

In addition, Dahiya et al. in their recent publications [48,49] discuss only one site for the interaction of OPs with albumin, claiming that this is Sudlow site I based on the evidence that warfarin (a specific ligand of Sudlow I) alters the efficiency of PM and TCPy interactions with albumin. However, a comparative analysis of the results of molecular docking and the crystal structure of albumin-warfarin complex (code 2bxd [23]) was not carried out and the stability of the obtained complex was not tested. Warfarin binds in subdomain IIA, at a distance of more than 15 angstroms from the site under consideration [48,49]. There could be allosteric effects there, but one cannot exclude the possibility of PM and TCPy binding at the site where warfarin binds (near Tyr149 of BSA). Moreover, different sites of interaction of structurally related methyl parathion are described: Trp214 (Trp213) of HSA (BSA) in the paper of Silva et al. [50] and Trp214 and Tyr411 of HSA in the study of Zhao et al. [51]. In our earlier publications, we assumed that Tyr150 and His242 of HSA (Tyr149 and His 241 in BSA) are the key participants in true esterase hydrolysis of OPs with albumin [13,14,26].

And finally, one of our principal notes is that OPs cannot be detoxified by true hydrolytic activity of albumin, as speculated by Dahiya et al. [48]. In the interaction of OPs with Sudlow site I of albumin, the first stage of binding and formation of the Michaelis-Menten complex without the formation of a covalent bond is principally important [6,26], which leads to not the detoxication but rather to preservation of OPs and its safer delivery to the main targets (synaptic acetylcholinesterase). That is, from the toxicological point of view, the consequences of such interaction are not positive, but exclusively negative, and to change it for better it is necessary to seek for modulators of the binding activity of albumin, which could weaken the interaction of OPs with Sudlow site I. In this case, OPs molecules would remain in the bloodstream for a longer time and could really be detoxified stoichiometrically or catalytically by other blood plasma esterases [52].

Considering the analysis of our and literature data, it can be assumed that, of the two main sites interacting with NPA, Sudlow I site with catalytic Tyr149 (Tyr150) exhibits true esterase activity, while Sudlow site II with catalytic Tyr410 (Tyr411) exhibits pseudo-esterase activity. Our long-term studies on pseudo (esterase) activity of albumin made it possible to propose a kinetic scheme for the interaction of the protein with a model substrate NPA (Figure 1) [6]. 

The work of each site (both the site of pseudo-esterase and true esterase activity) can be described by this scheme. At the first stage of interaction, the substrate binds to a site with the formation of an enzyme-substrate complex ES, then *p*-nitrophenol (P_1_) is released and the enzyme is acetylated (EA). The last step is deacylation of albumin and release of acetate (P_2_). The difference between pseudo-esterase and true esterase activity resides only in the lifetime of the acetylated EA enzyme, i.e. the difference is in the constant k_3_, which characterises the deacetylation reaction. The half-life of acetylated Tyr411, which is responsible for pseudo-esterase activity, is 61 ± 4 h [5], and in the case of true esterase catalysis the EA complex is in all likelihood a short-lived tetrahedral intermediate, similar to that formed in the process of hydrolysis of acetylcholine by acetylcholinesterase [42]. Moreover, if we neglect the short lifetime of the acetylated intermediate of true esterase activity and neglect the insignificant rate of Tyr411 deacetylation, then the proposed scheme can be used to describe the entire hydrolytic activity of albumin at two sites. At the first stage of the interaction of albumin with the substrate, it is adsorbed in Sudlow site II (ES), and then a rapid release of *p*-nitrophenol (P_1_) and acetylation of tyrosine Tyr411 (EA) occurs. At the second stage, the substrate binds to the site of esterase activity of albumin, where it is hydrolysed to acetate (P_2_) and *p*-nitrophenol (P_1_).

Back in 1986, concern was expressed that the classification of esterases did not take into account the facts that had accumulated by that time. Moreover, it was albumin that was cited as an example of a protein that exhibits esterase activity, but does not appear in the existing Enzyme Classification [53]. Our and other works of recent years emphasise the importance of assessing the enzymatic activity of albumin for the purposes of pharmaco- and toxicokinetics [4,6,54]. Quantitative analysis of the true esterase activity of albumin will help find a place for albumin in the enzyme nomenclature.

## 4. Materials and Methods

### 4.1. Materials

The following commercially available reagents were used: fatty acids free BSA (Sigma Aldrich, Steinheim, Germany), NPA (Sigma Aldrich, Steinheim, Germany), sodium trimethylsilylpropanesulfonate (DSS, Sigma Aldrich, Steinheim, Germany), *p*-nitrophenol (Acros Organics, Geel, Belgium), propionic acid (Chemical Line, St. Petersburg, Russia), thionyl chloride (Vekton, St. Petersburg, Russia). Deuterated dimethyl sulfoxide (DMSO d6) and deuterated water were provided by the Centre for Magnetic Resonance of St. Petersburg State University. NPP was synthesised from *p*-nitrophenol and propionic acid using thionyl chloride in the Research Institute of Hygiene, Occupational Pathology and Human Ecology, Federal Medical Biological Agency; according to NMR analysis, NPP content was not less than 90%.

### 4.2. NMR Spectroscopy

BSA was dissolved in phosphate buffered-saline (PBS) and deuterated water (9:1), NPA and NPP were dissolved in DMSO d6, and DSS was dissolved in double-distilled water. The reaction mixture for recording the hydrolysis of NPA and NPP by BSA was prepared by mixing BSA and DSS solutions, then, immediately before the spectrum was recorded, a substrate solution was added. In the final reaction mixture, BSA concentration was 360μM, DSS concentration was 1 mM, and substrate concentration was 7.2 mM.

NMR measurements were performed in the Centre for Magnetic Resonance, St. Petersburg State University Research Park. The prepared samples were scanned by one-dimensional ^1^H NMR on a Bruker Avance III 500 NMR spectrometer (Bruker, Karlsruhe, Germany) at room temperature (298K). Water suppression method using excitation sculpting with gradients was used. The residual peak of solvent in ^1^H spectrum (δ 4.7 ppm) was used as reference. Spectra were recorded every 3 min applying 8 scans for BSA-NPA reaction and 4 scans for BSA-NPP reaction. The relaxation delay (D1) of 15 s was applied for BSA-NPA interaction and 10 s for BSA-NPP interaction.

^1^H resonance lines of NPA, NPP and the products of their hydrolysis were assigned based on the spectra of spontaneous hydrolysis of the ligands (Appendix A) and multiplicity of the peaks (triplet for atom a_1_ and quadruplet for atom a_2_ of NPP (Appendix A)). To assign the signals of aromatic hydrogens, ^1^H NMR spectrum of *p*-nitrophenol was used [55]. 

### 4.3. Preparation of BSA and NPA Three-Dimensional Models

3D models of NPA, NPP and free cysteine were constructed and optimised by the steepest descent method using HyperChem program (version 8.0.8, Hypercube Inc., Gainesville, FL, USA) [56]. Crystal structure of BSA (PDB code 4jk4 [2]) was used as a three-dimensional model of reduced albumin. Water molecules and ligands were removed from the structure, and then the structure was optimised by energy minimisation using Chiron online service (Dokholyan laboratory, Penn State College of Medicine, PA, USA) [57].

The model of cysBSA was constructed as follows. Based on the available information on atomic charges, bond lengths, bond and torsion angles for different types of atoms and atomic groups, presented in the database of GROMACS 2018.1 software (University of Groningen, the Netherlands) [58], the topology of the cysteine amino acid residue bound through a disulfide bond with free cysteine was described. Then molecular docking of the free cysteine molecule into the redox site (Cys34) of BSA was performed. Based on the prepared topology and the coordinates of the obtained BSA-Cys complex, the GROMACS program generated the structure of the cysteinylated BSA, which was then optimised by the energy minimisation method.

### 4.4. Molecular Docking

At the next step, the prepared 3D model of reduced albumin was used for molecular docking of NPA and NPP into ligand-binding sites of BSA. Since the structures of the binding sites in the prepared models of reduced BSA and cysBSA are identical (changes in the structure affected only Cys34 and its environment), molecular docking of NPA into cysBSA was not performed. To start MD simulation of the complexes of cysBSA with NPA, we used the coordinates of the ligands that were obtained by molecular docking into reduced BSA.

Docking of NPA and NPP into albumin binding sites, as well as docking of free cysteine into the redox site of the protein, was performed using Autodock Vina 1.1.2 software package (The Scripps Research Institute, La Jolla, CA, USA) [59] and the online version of Rosetta software (Johns Hopkins University, Baltimore, MD, USA, http://rosie.rosettacommons.org, accessed on 26 September 2021) [60].

When Autodock Vina 1.1.2 was used, a search area of 15 × 15 × 15 Å^3^ was set in the studied protein binding site. The parameter called “exhaustiveness” characterising the amount of computational effort was set to 10. The parameter “energy_range” characterising the maximum scatter of energy values of conformations in the output file was set to 3 kcal/mol. The number of the most optimal conformations in the output file (num_modes) was set to 10. The result of the docking procedure was a set of 10 most probable conformations. 

When using the Rosetta program, the search area was set with a radius of 7 Å (maximum radius to search from starting coordinate); for other search parameters, default values were used. The number of structures to generate was set to 200. Generated structures were ranked by binding energy and by the energy contribution per residue (3 Å radius). Ligand-protein residue pairs were then ranked based on total energy contribution and orientation-dependent hydrogen bonding. Iterative refinement yielded the ten best scoring structures predicted for each ligand.

From the final set of the most probable conformations, we selected the conformation of the protein-ligand complex with a minimum distance between functionally significant atoms. In the case of free cysteine docking, this is the distance between the sulfur atoms of the ligand and Cys34; in the case of NPA and NPP, this is the distance between the carbonyl carbon of the ligand and the oxygen atom of the hydroxyl group of the catalytic amino acid.

### 4.5. Molecular Dynamics Simulation

Conformational changes in NPA-BSA complexes were calculated by the MD method using the GROMACS 2018.1 software and GROMOS53a6 force field [58]. The complexes were placed virtually into a cubic periodic box filled with water molecules. The single point charge (SPC) model was used to describe water molecules [61]. To neutralise the system, sodium ions were added. Temperature (300 K) and pressure (1 bar) were kept constant using a V-rescale thermostat [62] and a Berendsen barostat [63], with coupling constants of 0.1 ps and 1.0 ps, respectively. Long-range electrostatic interactions were treated by the particle-mesh Ewald method [64]. Lennard-Jones interactions were calculated with a cut off of 1.0 nm. The LINCS algorithm was used to constrain bonds length [65]. Before running the MD simulations, all the structures were minimised by steepest descent energy minimisation and equilibrated under NVT (1000 ps) and NPT (1000 ps) ensembles. The time step for MD simulation was 0.002 ps.

## 5. Conclusions

In this work, using the ^1^H NMR technique, we have demonstrated that serum albumin possesses true esterase activity. We have shown that the rate of formation of the products of hydrolysis of NPA and NPP in the presence of BSA is higher than in the case of spontaneous hydrolysis. The formation of the products of true esterase activity of BSA (acetate in the case of NPA and propionate in the case of NPP) begins no later than 10 min after the addition of the substrate solution to the protein solution. It is known that the half-life of acetylated tyrosine in the main site of pseudo-esterase activity of albumin is 61 ± 4 h. For other sites of acetylation, the formation of adducts with acetate requires at least 30 min. The combination of these facts indicates that there is a site of true esterase activity in the albumin molecule, which is different from the site of pseudo-esterase activity. Using molecular modelling methods, we have shown that Sudlow site I of BSA with catalytic Tyr149 (Tyr150 in HSA) is the most likely candidate for the role of the esterase center of albumin. The data obtained are of fundamental interest from the point of view of the evolution of the mechanism of enzyme functioning and practical value, since albumin interacts with many drugs and toxic compounds, largely determining their pharmacokinetics and toxicokinetics. Further quantitative analysis of the kinetic characteristics of the esterase activity of albumin will reveal details of the mechanism of functioning of this universal protein.

## Data Availability

The data presented in this study are available on request.

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
