# Peer review of "Esterase Activity of Serum Albumin Studied by 1H NMR Spectroscopy and Molecular Modelling"

_ijms, 2021, doi:10.3390/ijms221910593_

Round 1

Reviewer 1 Report

The authors show a significant esterase activity of albumin detected by 1H NMR. To my opinion, the signal to noise ratio is not very convincing. I am also missing any citation of faster NMR methods, for example rapid injection NMR, which possibly could help to improve the observation of the reaction at the beginning. Nevertheless, I agree with the publication of this paper.

Author Response

We are grateful to Reviewer-1 for the valuable comments on the manuscript. We have substantially modified the manuscript to address the points raised. The changes are detailed below and highlighted in the revised and improved manuscript.

To my opinion, the signal to noise ratio is not very convincing.

Section 2.7 has been expanded:

According to our data, BSA hydrolyses NPP clearly slower than NPA (Figures 1 and 8). However, for quantitative assessment of kinetic constants using NMR, the slow reaction rate is rather an advantage with respect to duration of scanning the sample at one time point. For the same reason, we used a reduced number of scans in order to find a compromise between the signal-to-noise ratio and the error in determining the hydrolysis rate due to prolonged scanning duration. A quantitative analysis of true esterase activity of albumin is the goal of our next studies.

I am also missing any citation of faster NMR methods, for example rapid injection NMR, which possibly could help to improve the observation of the reaction at the beginning.

Section 2.1 has been expanded:

Due to the features of NMR technology, usually 7-10 minutes elapse between the moments of mixing the substrate with the enzyme and recording of the first spectrum. By this time, Sudlow sites II of BSA molecules are already completely acetylated, therefore the obtained spectra reflect the kinetics of exclusively true esterase reaction of albumin. That is why such a delay is applicable for measurement of kinetics of albumin true esterase activity, however, to study the reaction of Sudlow site II (which occurs during the first 5 minutes of BSA-NPA interaction), faster NMR methods should be applied [Schanda, 2007; Arthanari et al., 2019]. For example, rapid injection NMR spectroscopy allow the observation of fast chemical reactions and unstable intermediates [Denmark et al., 2010; Jensen et al., 2019].

Arthanari, H.; Takeuchi, K.; Dubey, A.; Wagner, G. Emerging solution NMR methods to illuminate the structural and dynamic properties of proteins. Curr. Opin. Struct. Biol. 2019, 58, 294-304. doi: 10.1016/j.sbi.2019.06.005.

Denmark, S.E.; Williams, B.J.; Eklov, B.M.; Pham, S.M.; Beutner, G.L. Design, validation, and implementation of a rapid-injection NMR system. J. Org. Chem. 2010, 75(16), 5558-5572. doi: 10.1021/jo100837a.

Jensen, P.R.; Matos, M.R.A.; Sonnenschein, N.; Meier, S. Combined In-Cell NMR and Simulation Approach to Probe Redox-Dependent Pathway Control. Anal. Chem. 2019, 91(8), 5395-5402. doi: 10.1021/acs.analchem.9b00660.

Schanda, P. Development and application of fast NMR methods for the study of protein structure and dynamics. Biological Physics [physics.bio-ph].  Université Joseph-Fourier, Grenoble I, 2007. Available online: https://tel.archives-ouvertes.fr/tel-00181457/document (accessed on 20 September 2021).

Reviewer 2 Report

Belinskaia et al applied 1H-NMR and molecular modelling to study the esterase activity of bovine serum albumin. They postulated that Sudlow site I is the site of true esterase activity. Although 1H-NMR methods were applied to detect the acetate product formation due to the BSA activity, the fact that the study was more qualitative than quantitative made the real novelty of the paper come from the computational study. Since similar studies were done previously, I found it hard, with the present discussion, to attest to the real novelty. Therefore, I do believe the authors should enhance the overall discussion to better point it for the non-specialist reader.

1) Although the authors claimed that they could not find a report applying NMR spectroscopy to study the enzymatic activity of albumin, they do exist. Dahiya et al observed time-dependent changes in the 1H NMR intensity of PM in the presence of BSA, suggesting faster degradation of PM with increasing protein concentration (10.1016/j.pestbp.2019.10.004). Keishin et al also applied NMR to observe the hydrolysis rates of acetylsalicylate in the presence of HSA (10.1016/0161-5890(91)90093-Y). The authors should include these other studies in their discussion and better point the benefits of their applied methodology.

2) Since for the NMR experiments the protein concentration is quite high (360 µM) and the authors claimed in the methods having used the “fatty acids free BSA (Sigma Aldrich)”, it is not clear to me which sample it is and their overall purity. Since there are no quantitative analyses, could another esterase come as an impurity in the sample account for the acetate formation? 

3) In the abstract of their previous published manuscript (Bioorg Khim. Sep-Oct 2014;40(5):541-9 - that I could not find an English version) they had already proposed the true esterase activity of albumin Tyr150 (BSA Tyr149). Some of the authors also proposed it previously in doi:10.3390/molecules22071201. Besides, a detailed QM/MM study on the esterase activity of Sudlow site 1 was published and not discussed in this manuscript (https://pubs.acs.org/doi/pdf/10.1021/jp506629y). Since I am not a specialist in the field of BSA esterase activity, I found it difficult to point the real novelty of these finds in the present work. The authors might point this better for the non-specialist reader.  

4) In their discussion of Sudlow site 1 being more susceptible to allosteric modulation, the architecture of this site remained unaltered in the cystenylated BSA. Since it is clear from the MD data that a disturbance in the BSA/NPA interaction occurred, the authors should better explore this from the BSA structural point of view. In the simulation of ligand-free cystenylated BSA, is there any changes in the structural dynamic or the web of interactions inside the Sudlow I? Can the authors give further structural/dynamic details on this allosteric modulation?

Minors:

1) The reference (or either methodology) for the 1H resonance assignment of NPA and p-nitrophenyl propionate should be included.

2) Based on their interest in exploring the specific amino acids playing a role in enzymatic catalyses and protein-ligand interaction, future analyses including frustrations studies might be useful. The authors can check for http://frustratometer.qb.fcen.uba.ar/   (10.1093/bioinformatics/btab176)

Round 2

Reviewer 2 Report

The authors have satisfactorily responded to all my questions and made the necessary changes to the manuscript
